# High-Throughput Determination of Multiclass Chemical Hazards in Poultry Muscles and Eggs Using UPLC–MS/MS

**DOI:** 10.3390/foods14101660

**Published:** 2025-05-08

**Authors:** Rong Chen, Lan Chen, Mingyue Du, Qiaozhen Guo, Ciping Zhong, Jing Zhang, Xiaoqin Yu

**Affiliations:** 1Key Laboratory of Liquor Regulation Technology, State Administration for Market Regulation, Sichuan Institute of Food Inspection, No. 8 Xinwen Road, West Hi-Tech Zone, Chengdu 611731, China; rong20221212@163.com (R.C.); 17713637579@163.com (C.Z.); 2Beijing Key Laboratory of Diagnostic and Traceability Technologies for Food Poisoning, Beijing Center for Disease Control and Prevention, Beijing 100013, China; cdsj2021@163.com (L.C.); dumy24818@163.com (M.D.); qiao037@126.com (Q.G.); 3Irradiation Preservation Key Laboratory of Sichuan Province, Chengdu Institute of Food Inspection, No. 10, Section 2, Furong Avenue, Wenjiang District, Chengdu 611130, China

**Keywords:** poultry products, chemical hazards, high-throughput detection, UPLC–MS/MS

## Abstract

A high-throughput method for the determination of a variety of chemical hazards in poultry muscle and egg samples was established via ultra-performance liquid chromatography–tandem triple quadrupole mass spectrometry (UPLC–QqQ-MS). The sample preparation procedure was developed based on this quick, easy, cheap, effective, rugged, and safe (QuEChERS) method and validated for 280 chemical hazards potentially present in poultry products. The target compounds in poultry samples were extracted with a 1% formic acid–acetonitrile solution (15:85, *v*/*v*), and the metal ions in the matrix were chelated by adding ethylenediaminetetraacetic acid disodium salt (Na_2_EDTA). The supernatant was purified using Enhanced Matrix Removal (EMR) lipid sorbent. Chromatographic gradient separation was performed on an ACQUITY UPLC BEH C18 (2.1 mm × 100 mm, 1.7 μm) column with multiple reaction monitoring (MRM) under both negative- and positive-ion mode. Internal standard calibration or matrix-matched calibration was used for the quantitation. The results showed that good linearity was achieved for each target compound with correlation coefficients (R^2^) ≥ 0.99. The limits of detection (LODs) ranged from 0.05 to 10 µg/kg, and the acceptable limits of quantification (LOQs) were determined to be 0.1–20 µg/kg for all 280 compounds. Approximately 90% of the target compounds exhibited mean recoveries ranging from 60% to 120%, with relative standard deviations (RSDs) within 16.2%. This method can be used for the high-throughput rapid detection of prohibited drug residues in poultry eggs due to its easy operation and high accuracy. It was applied in real sample detection, and 43 chemicals including metronidazole were found in 211 poultry samples, with a concentration range of 0.11–638 μg/kg.

## 1. Introduction

Poultry products, including muscle and eggs, are common animal-derived foods in daily life due to being rich in protein, vitamins, minerals, and other nutrients. China is a major producer and consumer of poultry products. According to the report of the National Bureau of Statistics [1], the off-take volume nationwide of poultry in 2024 was 17.3 billion, which means that the quality of these products is closely related to consumer health and national interests. Poultry product safety has become a hot topic in recent years. The government has established standards to regulate the breeding industry. However, during poultry farming, the abuse or non-compliant use of veterinary drugs poses a risk of excessive residues in the final products (including, but not limited to, sulfonamides, quinolones, tetracyclines, and β-agonists). Additionally, the environmental pollution produced by farming sites and feed can lead to the pollution or accumulation of contaminants through the food chain, thereby threatening consumer health. The long-term consumption of products with excessive levels of chemical residues may cause chronic toxicological effects, such as drug resistance; allergic reactions; and even teratogenic, carcinogenic, or mutagenic effects [2,3]. Therefore, it is essential to establish a high-throughput detection method to monitor multiclass chemical hazards in poultry products.

The current methods for drug residue analysis in animal-derived foods mainly include gas chromatography [4,5], high-performance liquid chromatography [6,7], and liquid chromatography–tandem mass spectrometry (LC–MS/MS). LC–MS/MS is a common method for analyzing drug residues in animal-derived foods. It is characterized by high selectivity and sensitivity and low detection limits [8,9,10,11,12]. The widely applied pre-treatment methods for drug residue analysis in animal-derived foods mainly include solid-phase extraction [13,14,15] and the QuEChERS approach [16,17]. The former method requires enrichment, separation, and purification steps, which are costly and time-consuming; meanwhile, solid-phase extraction columns are mostly based on the principles of polar separation and ion-exchange separation, so they are not suitable for the analysis of a large number of compounds with widely varying properties [18,19]. In contrast, the QuEChERS method, known for its rapidity, low cost, and simplicity, is widely used for drug residue analysis in animal-derived foods and enables the high-throughput analysis of compounds with significantly different properties [20,21]. Therefore, this study aims to establish a QuEChERS ultra-performance liquid chromatography–tandem triple quadrupole mass spectrometry (UPLC–MS/MS) method for detecting 280 chemical hazards in poultry products, which will provide technical support for monitoring chemical residues and ensuring food safety.

## 2. Materials and Methods

### 2.1. Materials and Reagents

A total of 280 analytical standards and 44 isotope-labeled internal standards (listed in Appendix A), all with purity > 98%, were obtained from TRC (Toronto, ON, Canada) or Alta Scientific Co., Ltd. (Tianjin, China). Zirconia-bonded octadecyl silica (Z-sep/C18), octadecyl-bonded silica (C18), and N-propylethylenediamine-bonded solid phase (PSA) adsorbent materials were obtained from SUPELCO (Bellefonte, PA, USA); multiwalled carbon nanotube (MWNT, 8–15 nm) adsorbent materials were purchased from Xianfeng Nanomaterials Technology (Nanjing, China); and QuEChERS dSPE EMR-lipid purification packing material was purchased from Agilent (Santa Clara, CA, USA).

HPLC-grade acetonitrile and methanol were obtained from Dima Technology Ltd. (Beijing, China); formic acid was purchased from Acros Corporation (Lexington, MA, USA) with an available purity > 95%; disodium EDTA (Na_2_EDTA), sodium chloride, anhydrous sodium sulfate, and ammonia were purchased from Sinopharm Chemical Reagent Co. (Beijing, China); and ammonium fluoride (purity > 99%) was purchased from Aladdin Biochemical Technology Co., Ltd. (Shanghai, China). Ultrapure water was prepared using a Milli-Q system (Millipore, MA, USA).

### 2.2. Instruments and Equipment

The instruments and equipment used in this study were as follows: an ACQUITY^TM^ Ultra Performance Liquid Chromatograph—Xevo^®^ TQ-XS Tandem Mass Spectrometer (Waters, Milford, MA, USA); an Allegra X-30R Centrifuge (Beckman, Brea, CA, USA); a Vortex-Genie 2 Vortex Oscillator (Scientific Industries, Bohemia, NY, USA); and a Mettler Toledo XPE 105 Electronic Balance (Greifensee, Switzerland).

### 2.3. Sample Preparation

Blank samples of chicken muscle and eggs used for method development or validation were kindly donated by China Agricultural University, while the samples for detection are collected from local markets in Beijing and Ningxia.

Poultry muscle samples were prepared by removing connective tissues, dicing into small pieces, and homogenizing using a mechanical grinder. Egg samples were prepared by pooling and homogenizing 10 whole eggs. Each 2 g homogenized egg/muscle sample was accurately weighed out, placed in a 50 mL centrifuge tube, and spiked with 100 μL of 100 μg/L IS solution. Then, 1.5 mL of ultrapure water was added and vortexed for 30 s; following this, 8.5 mL of acetonitrile, 100 μL of formic acid, and 25 mg of ethylenediaminetetraacetic acid disodium salt were added to the centrifuge tubes and mixed by vortexing for 1 min. The mixture was ultrasonically extracted for 30 min. Afterward, the samples were centrifuged at 10,000 r/min for 10 min at 4 °C. The supernatant was transferred into a new centrifuge tube, 2 g of anhydrous sodium sulfate and 1 g of sodium chloride solid were added [22], and the mixture was vortexed for 1 min and then centrifuged at 10,000 r/min for 5 min at 4 °C. For the extract cleanup, 1 mL of supernatant was then vortexed by adding 50 mg of dSPE EMR-lipid [23,24] sorbent and vortexing for 1 min. After centrifugation at 4 °C at 14,000 r/min for 5 min, the supernatant was pipetted into a glass vial and injected into the UPLC–MS/MS for analysis.

### 2.4. Instrumental Conditions

The analytical column was an ACQUITY UPLC BEH C18 (2.1 mm × 100 mm, 1.7 μm; Waters, Milford, MA, USA) with a flow rate of 0.3 mL/min at 40 °C. In positive-ion mode, the phase A was methanol-acetonitrile (*v*/*v*, 1:1), and phase B was a 0.5 mmol/L ammonium fluoride—0.1% formic acid aqueous solution, with the gradient elution procedure as follows: 0~2 min, 3% A; 2~5 min, 3~15% A; 5~10 min, 15% A; 10~15 min, 15~30% A; 15~20 min, 30~50% A; 20~24 min, 50~100% A; 24~27 min, 100% A; 27~28 min, 100% A; 28~28.5 min, 100%~3% A; 28.5~30 min, 3% A. In negative-ion mode, the mobile phase A was acetonitrile, and phase B was a 1 mmol/L ammonium fluoride aqueous solution; the elution gradient was as follows: 0~1 min, 10~15% A; 1~2.5 min, 15~50% A; 2.5~6 min, 50~65% A; 6~7.5 min, 65~95% A; 7.5~8 min, 95~100% A; 8~8.5 min, 100% A. The injection volume was 3 µL in positive-ion mode and 5 µL in negative-ion mode.

MS/MS analyses were performed on a TQ-XS triple quadrupole mass spectrometer with electrospray ionization (ESI) in multiple-reaction-monitoring (MRM) mode. The ESI settings were as follows: capillary voltage, 2.5 kV; cone voltage, 30 V; ion source temperature, 150 °C; dissolvent gas temperature, 400 °C; dissolvent gas (N_2_) flow rate, 800 L/h; pressure of collision chamber, 3.2 × 10^−3^ mbar. The compound-dependent parameters like precursor ions, product ions, and collision energies are listed in Appendix A.

### 2.5. Preparation of Standards and Working Solutions

Standard stock solutions at 1000 mg/L were prepared in methanol solution with 0.1% formic acid and stored at –20 °C prior to use. The standard working solutions were diluted with methanol to the desired concentration. A mixed IS solution of 44 isotope-labeled chemicals (Appendix A) was diluted to 100 μg/L.

### 2.6. Quality Assurance

A robust quality assurance design for LC–MS/MS ensures data integrity, regulatory compliance, and reliable results. In this study, the method was validated according to China National Standard GB/T 27417-2017 Conformity assessment—Guidance on validation and verification of chemical analytical methods [25]. Various parameters including linearity, accuracy, precision, limit of detection (LOD), and limit of quantification (LOQ) were evaluated during the course.

The performance of the LC–MS/MS analytical system was verified using the evaluated standard solutions, including the analytes and internal standards. With the calibration curve established, accuracy should be +15% for all points except the lowest calibrator (+20%). In addition, the curve slope should be r^2^ > 0.99. A spiked blank matrix sample was used as the quality control (QC) sample due to the lack of certified reference materials. A blank sample, as well as QC samples, was applied for every 20 injections in a batch analysis.

## 3. Results and Discussion

### 3.1. Optimization of UPLC–MS/MS

#### 3.1.1. Mass Spectrometry Parameters

The mass spectrometry parameters were optimized for individual standards. A full scan of the primary mass spectrum was performed in positive- and negative-ion modes. Ultimately, satisfactory results were achieved for 262 substances in the positive-ion mode, while the remaining 18 substances performed better in the negative-ion mode. The cone voltage was determined by optimizing the response of the precursor ion; then, a product ion scan of the secondary mass spectrum was performed, the fragment ions with the highest response were selected as the quantitative ions, and the second-highest fragment ions were selected as the qualitative ions by changing the collision energy. The detailed mass spectrometry parameters are shown in Appendix A.

#### 3.1.2. Chromatographic Conditions

The mobile phase composition was systematically optimized to achieve optimal response and separation for all 280 target compounds. Three organic phase compositions were evaluated: methanol, acetonitrile, and methanol-acetonitrile (1:1, *v*/*v*). Concurrently, two aqueous phase formulations were compared for use in positive-ion mode: (1) 0.1% formic acid and (2) 0.5 mmol/L ammonium fluoride containing 0.1% formic acid. The results (Figure 1A,B) demonstrated that the addition of 0.5 mmol/L ammonium fluoride into a 0.1% formic acid aqueous solution significantly improved the chromatographic resolution and enhanced the mass spectrometric response intensities for most target analytes compared to using a formic acid solution alone. In organic phase screening, the methanol-acetonitrile mixed system exhibited superior comprehensive performance (response intensity and resolution), markedly outperforming individual solvent systems, and was, therefore, selected as the optimal organic phase.

Based on these findings, the final optimized mobile phase combination for positive-ion mode analysis was established as follows: methanol-acetonitrile (1:1, *v*/*v*) with a 0.5 mmol/L ammonium fluoride aqueous solution containing 0.1% formic acid.

Under the negative-ion mode (Figure 2), the aqueous phase was compared with three solutions, 1 mmol/L ammonium fluoride, water, and 0.01% ammonia, and it was seen that the response value was higher when the aqueous phase was ammonium fluoride and water. The organic phase was compared with methanol and acetonitrile. Finally, acetonitrile—1 mmol/L ammonium fluoride solution was used as the mobile phase in the negative analysis.

### 3.2. Optimization of Sample Pretreatment

#### 3.2.1. Extraction Solvents

Aqueous acetonitrile solution is the most commonly used extract in animal-derived food sample treatment due to its excellent protein precipitation effect and organic solubility. In this study, pure acetonitrile, 85% acetonitrile aqueous solution, and 50% acetonitrile aqueous solution were used for the chicken muscle and egg extraction. From a visual perspective (as shown in Appendix A), the supernatant of the 50% acetonitrile aqueous solution remained turbid even after 10 min of high-speed centrifugation. In contrast, when extracted with pure acetonitrile or 85% acetonitrile, the supernatant appeared significantly clearer. However, pure acetonitrile caused the tight aggregation of muscle samples, resulting in relatively poor extraction efficiencies for the majority of target compounds. We also compared the extraction results of an initial homogenization with 1.5 mL of water followed by the addition of 8.5 mL of acetonitrile. This approach demonstrated a clear supernatant and improved sample dispersion. Additionally, acidic conditions have been proven to enhance the extraction efficiency and response of many drugs [26]. Consequently, 0.1% formic acid was subsequently introduced into the sample treatment process. The results indicated that the addition of formic acid enhanced the response values for approximately 48% of the target compounds. In conclusion, a mixture of 1.5 mL of water, 8.5 mL of acetonitrile, and 0.1% formic acid was selected as the extraction solvent in this study.

#### 3.2.2. Purification Materials

Proteins, lipids, and other impurities in the sample extracts could interfere with the assay of the target compounds, and thus the extracts need further purification to reduce matrix interference before analysis. Currently, the dispersed purification materials commonly used in the QuEChERS approach include C18 [27,28], EMR [29,30], PSA [31,32], MWNTs [33], GCB [34], etc. In this experiment, the purification effects of five sorbents, namely, 50 mg of C18, 25 mg of PSA, 50 mg of Z-Sep/C18, 10 mg of MWNTs, and 50 mg of EMR-lipid, were compared. The dosage of different sorbents were determined upon the literature review or pre-experiment basis. In the chicken muscle matrix, the EMR-lipid sorbent showed optimal performance in terms of recovery rate, with 90% of the target compounds achieving recovery within the validated acceptable range (60–120%) and only 8% showing recovery below 60%, fulfilling all essential criteria for multiclass veterinary drug residue analysis. PSA ranked second, while MWNTs induced the worst recovery, with 49% of the targets achieving a recovery lower than 60%. Accordingly, the matrix effect (ME, Figure 3) of many targets was significantly weakened after the purification; among them, there were 14 targets with matrix inhibition in the PSA groups and 65 targets in the EMR-lipid group (ME < 80%), and the percentage of compounds with matrix inhibition in these two groups was significantly lower than that in other groups. The target recovery rates of EMR-lipid and PSA were further compared by plotting the 29 substances with the largest differences in recovery rates when using these two sorbents. Figure 4 demonstrates that the recovery rates of the 29 compounds through EMR-lipid purification were higher than those achieved with PSA. This difference can be attributed to the fact that EMR-lipid filler is a lipid-enhanced sorbent, which effectively removes impurities such as phospholipids in meat without compromising the recovery of target compounds. In contrast, PSA filler eliminates organic acids and other matrix impurities, and it is commonly used for pesticide residue analysis. However, in this experiment, PSA exhibited a matrix-enhancing effect (ME > 120%) and resulted in lower recoveries for many target compounds [35,36,37]. With respect to the recovery rate and matrix effect, EMR-lipid sorbent was selected for the chicken muscle purification. Similar comparisons were conducted for the treatment of egg extracts, and EMR-lipid sorbent was also proven to be the most effective option. Figure 5 and Figure 6 present statistical comparisons of target compounds demonstrating > 20% recovery differences between the two optimal purification sorbents (EMR-lipid and PSA) in chicken muscle and (EMR-lipid and Z-Sep/C18) egg matrices, respectively.

### 3.3. Method Validation

#### 3.3.1. Matrix Effect

The matrix effect (ME) of the target compounds was evaluated by calculating the ratio of the slope of the matrix-matched standard curve to the slope of the solvent-based standard curve. A ratio within the range of 0.80–1.20 was considered negligible, while values greater than 1.20 indicated matrix enhancement and values less than 0.80 indicated matrix inhibition. As shown in Figure 7, the matrix effects for egg samples ranged from 0.53 to 1.44, with 54 target compounds exhibiting matrix inhibition. Among these, 11 targets, including Sudan 2, demonstrated a matrix effect of less than 0.60. Similarly, for chicken muscle, the matrix effects ranged from 0.54 to 1.49, with 62 target compounds showing matrix inhibition, of which 8 targets, including clenbuterol, had a matrix effect below 0.60.

#### 3.3.2. Linear Range and Method Limit of Quantification (LOQ)

IS-spiked standard calibration curves or matrix-matched calibration curves were prepared using matrix blank extracts with concentrations of 0.04, 0.1, 0.2, 0.4, 1, 2, 4, 10, 20, 40, and 100 μg/L (with a constant IS concentration of 1 μg/L). These samples were then analyzed under the optimized conditions described above. Linear regression analysis was performed using the mass concentration of the target compounds as the independent variable (*x*-axis, μg/L) and the corresponding peak area ratios of the external standards to the IS as the dependent variable (*y*-axis). As shown in Appendix A in the Appendix A, the target showed a good linear relationship between mass concentration and peak area with a correlation coefficient R^2^ ≥ 0.99 within the linear range of 0.04~100 μg/L. The method demonstrated excellent sensitivity, with limits of detection (LODs, S/N ≥ 3) ranging from 0.05 to 10 μg/kg and limits of quantification (LOQs, S/N ≥ 10) ranging from 0.1 to 20 μg/kg. In total, 228 target compounds exhibited high sensitivity with LOQs between 0.1 and 2 μg/kg, while only 8 compounds had LOQs exceeding 10 μg/kg. Compared to the published literature focused on multiresidue detection in eggs and muscles, this study presented comparable or higher sensitivity (Table 1).

#### 3.3.3. Spiking Recoveries and Precisions

Spiking experiments were conducted at three concentration levels (1 μg/kg, 10 μg/kg, and 50 μg/kg) using blank egg and chicken muscle samples. For each concentration level, six replicate groups were prepared. The results demonstrated that the recoveries of the target compounds in the egg matrix ranged from 55.4% to 139%, with relative standard deviations (RSDs) ranging from 1.14% to 16.2%. Similarly, in the chicken matrix, the average recoveries ranged from 57.9% to 137%, with RSDs between 0.98% and 15.9%. It needs to be noted that 253 of the target compounds exhibited mean recoveries ranging from 60% to 120% at the three spiked levels. Figure 8 depicts the distribution of the recoveries, and it can be seen that the median recoveries of the targets at different concentration spiked levels ranged from 84.3% to 103%, which is acceptable and meets the requirements for the analysis of multidrug residues.

### 3.4. Application

This newly developed high-throughput method for chemical residue determination in poultry products was applied to 144 poultry egg and 67 chicken muscle samples. In total, 114 poultry eggs (eggs and egg products: 63 eggs, 1 goose egg, 1 pigeon egg, 1 quail egg, 9 salted duck eggs, 9 preserved eggs, and 9 soft-boiled eggs) and 37 poultry muscle samples were collected in February, May, and July 2021 from supermarkets and grocery in Beijing, while 30 chicken muscle and 30 egg samples were collected in July 2021 in Ningxia. A total of 43 compounds were detected in poultry egg samples, including sulfonamides, nitroimidazoles, quinolones, sedatives, anticoccidials, anti-inflammatories, and hormones. Since the hormone substances presented in Table 2 such as progesterone, 17α-hydroxyprogesterone, hydrocortisone, cortisone, corticosterone, androstenedione, and testosterone have been reported to be naturally present in mammals with different ranges of content at different stages of growth [44], in general, these endogenously present hormone substances pose no health risk to consumers at the detected levels. Therefore, hormones are not included in the description below.

A total of 17 targets were detected in 67 chicken muscle samples at concentrations ranging from 0.11 to 333 μg/kg, with the maximum value observed for toltrazuril. A total of 27 targets were found in 144 poultry egg samples at concentrations ranging from 0.11 to 638 μg/kg, with the maximum value observed for sulfadiazine (Table 2). It is worth noting that 12 target compounds were detected in 18 egg products, of which 6 target compounds were detected in 1 salted duck egg sample simultaneously (Figure 9 ), indicating that the supervision of egg products needs to be strengthened.

## 4. Conclusions

An UPLC–MS/MS method was established for the simultaneous determination of 280 chemical hazards that may be present in poultry samples. Through the optimization of extraction solvents and purification materials, acidic acetonitrile solution and the EMR-lipid filler were selected as the extraction and purification agent, respectively, to reduce the influence of interferences in the matrix. All the target analytes exhibited linear responses (R^2^ ≥ 0.99) within their respective calibration ranges. The spiked recoveries ranged from 55.4% to 139% in the egg matrix and 57.6% to 137% in the chicken matrix, and the limits of quantification (LOQs) were between 0.1 and 20 μg/kg. Compared with the other reported methods, this method covers more types of compounds and achieved the high-throughput detection of chemical hazards. It has been applied to the detection of over 200 actual samples. This method is suitable for the routine monitoring of chemical hazards in poultry products, providing technical support and a valuable reference for ensuring the quality and safety of these products. While this method enables broad compound coverage, certain analytes demonstrate limited recovery efficiency, and the reliance on costly EMR-lipid sorbents may hinder large-scale implementation. Future studies should prioritize advanced sorbent development to improve recovery and automation-integrated workflows to boost throughput.

## Figures and Tables

**Figure 1 foods-14-01660-f001:**
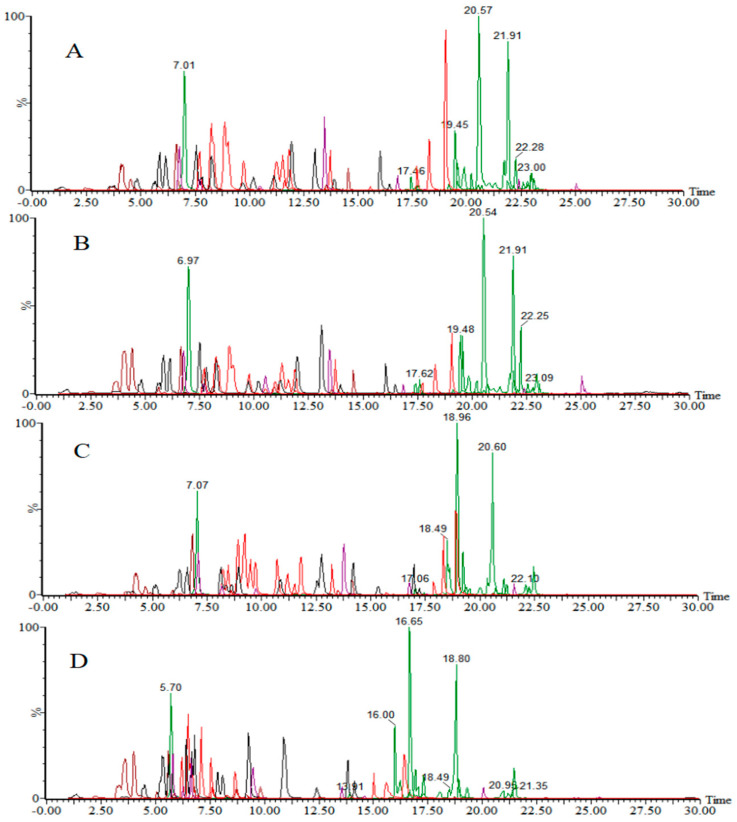
Total ion chromatograms of target compounds using different mobile phases in positive-ion mode: (**A**) methanol-acetonitrile (1:1, *v*/*v*)—0.5 mmol/L ammonium fluoride—0.1% formic acid in water; (**B**) methanol-acetonitrile (1:1, *v*/*v*)—0.1% formic acid in water; (**C**) acetonitrile—0.5 mmol/L ammonium fluoride—0.1% formic acid in water; (**D**) methanol—0.5 mmol/L ammonium fluoride—0.1% formic acid in water.

**Figure 2 foods-14-01660-f002:**
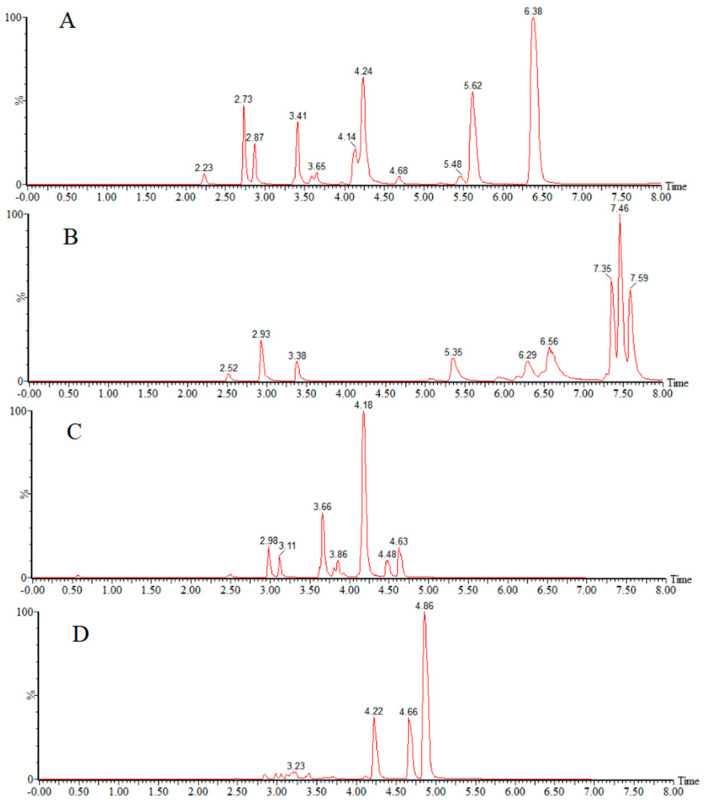
Total ion chromatograms of target compounds using different mobile phases in negative-ion mode: (**A**) acetonitrile—1 mmol/L ammonium fluoride; (**B**) methanol—1 mmol/L ammonium fluoride; (**C**) acetonitrile—0.01% ammonium hydroxide in water; (**D**) acetonitrile-water.

**Figure 3 foods-14-01660-f003:**
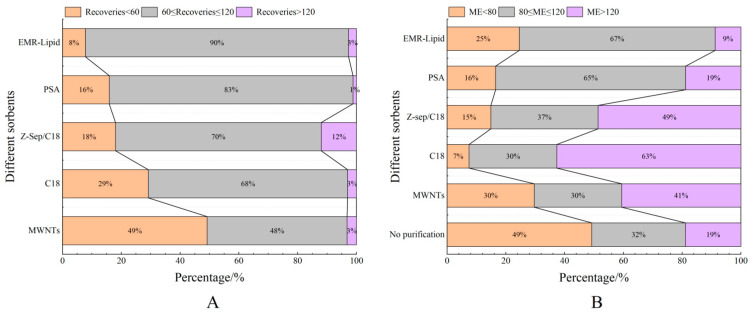
Distribution of recovery (**A**) and matrix effects (**B**) of target compounds when using different sorbents in chicken muscle.

**Figure 4 foods-14-01660-f004:**
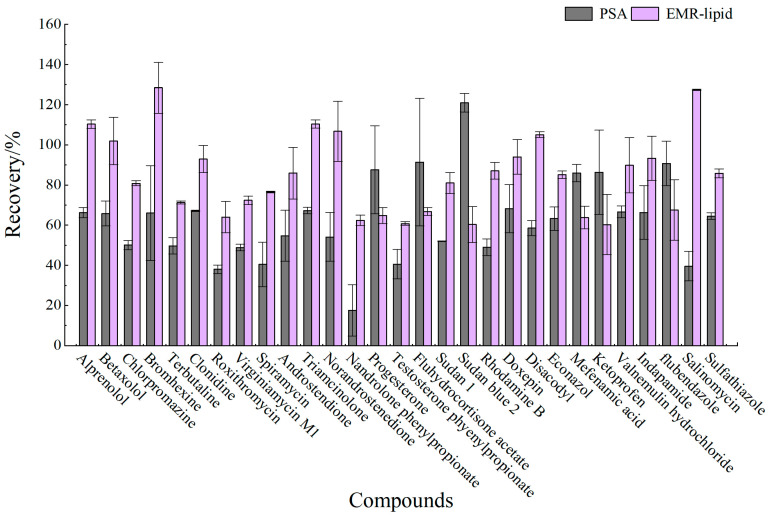
Recovery of 29 targets in chicken muscle using EMR-lipid and PSA purification.

**Figure 5 foods-14-01660-f005:**
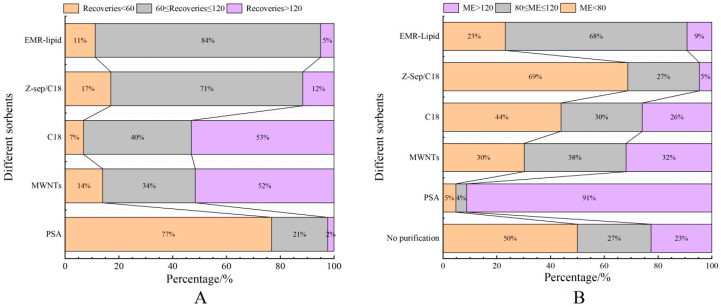
Distribution of recovery (**A**) and matrix effects (ME, (**B**)) of target compounds when using different sorbents in egg samples.

**Figure 6 foods-14-01660-f006:**
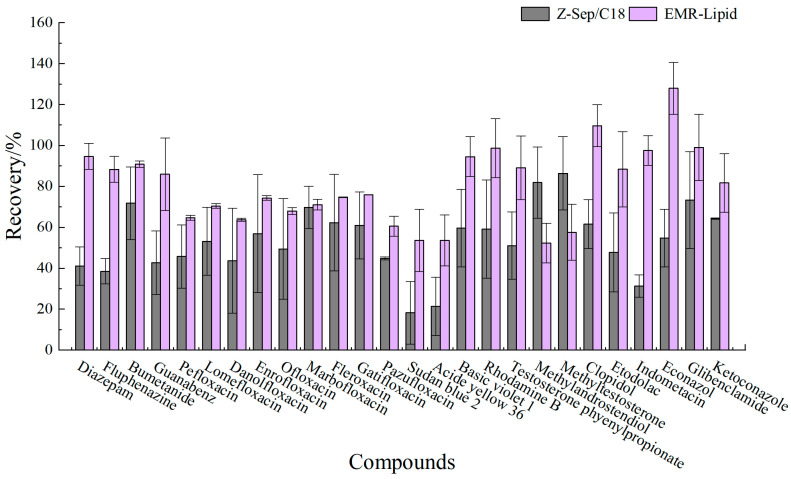
Recovery of 26 targets in eggs using EMR-lipid and Z-Sep/C18 purification.

**Figure 7 foods-14-01660-f007:**
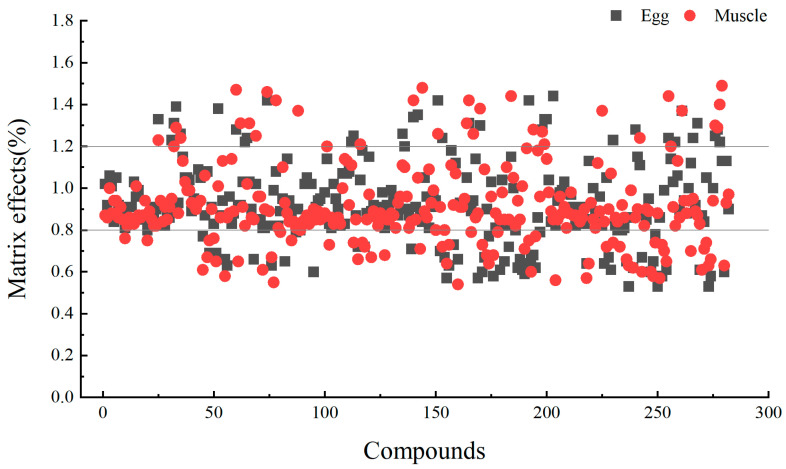
Matrix effects of 280 targets in chicken egg and chicken muscle (the compound number is the same as that in Appendix A).

**Figure 8 foods-14-01660-f008:**
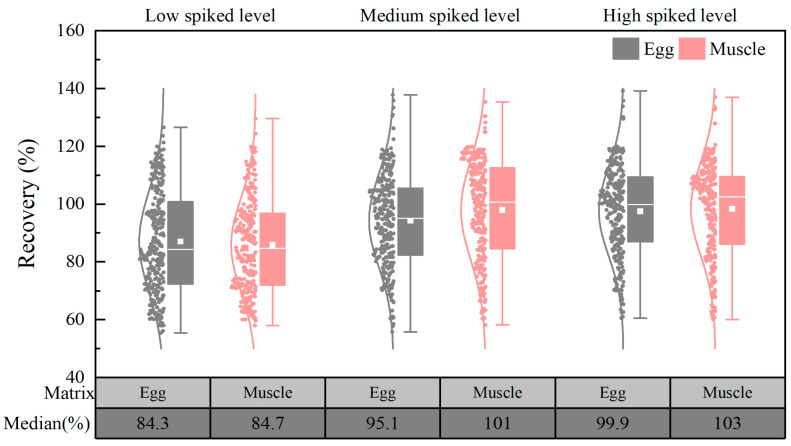
Distribution of target recovery in chicken and egg matrices (*n* = 6).

**Figure 9 foods-14-01660-f009:**
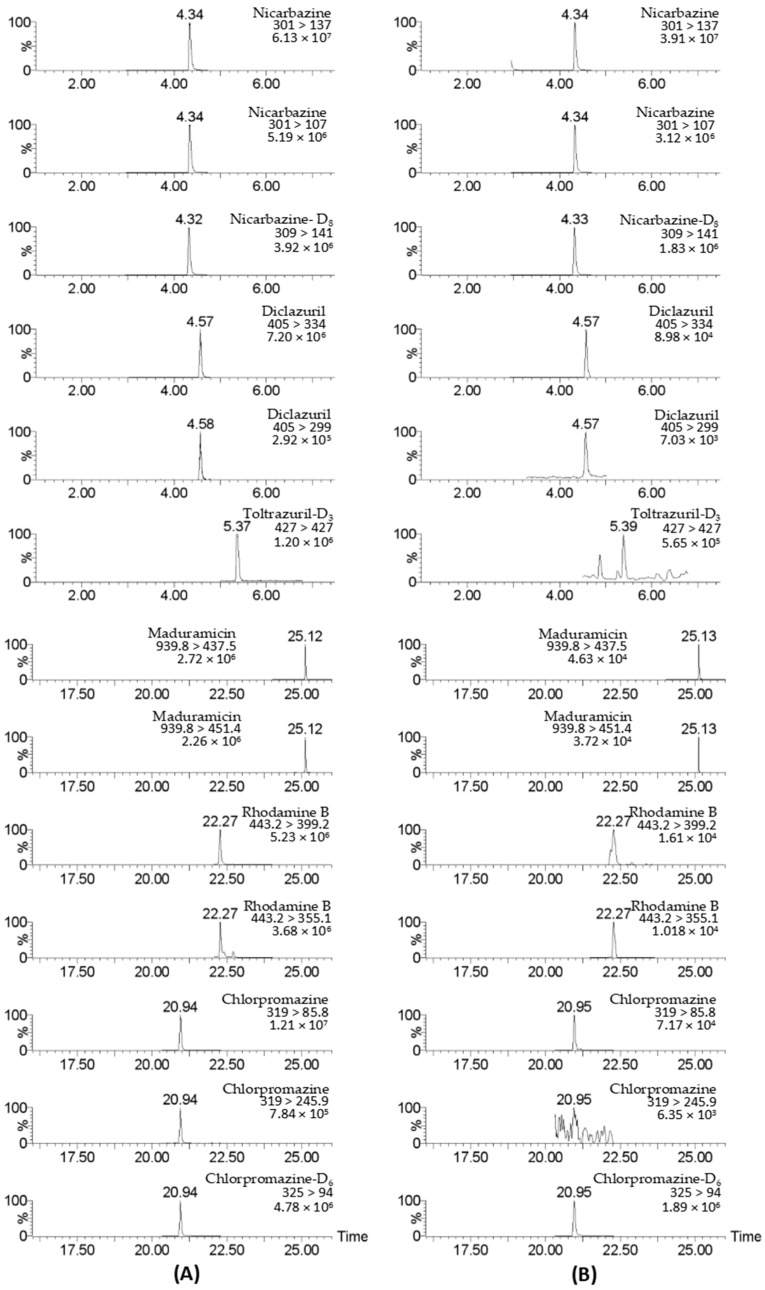
UPLC–MS/MS chromatograms of detected drugs in standard solution (**A**) and a salted duck egg sample (**B**). The standard concentration was 5 μg/L.

**Table 1 foods-14-01660-t001:** Summary of the literature on multidrug residues analysis in egg and chicken muscle matrices.

Matrix	Compound	MDL(μg/kg)	Detection Technique	Column	Time Cycle (min)	Pretreatment	Recovery (%)	References
Eggs	244 chemical contaminants	(LOQ) ≤ 5	LC–MS/M	Zorbax Eclipse XDB-C18	25.1	Extraction with 5% formic acid in acetonitrile; purification: EMR-lipid	51.33–118.22	[38]
Eggs	285 compounds (pesticides, veterinary drugs, and mycotoxins)	LOQ: 0.1–50	UHPLC–MS/MS	BEH C18	25	Extraction with 1% formic acid in acetonitrile	Satisfactory recoveries (70 to 120)	[39]
Chicken eggs	169 veterinary drugs	LOD: 0.01–3.81	UHPLC–MS/MS	Eclipse Plus C18	24	Extraction with ethyl formate–acetonitrile; purification: EMR-lipid dSPE	57–124	[40]
Eggs	78 veterinary drugs	LOQ: 0.1–1	UHPLC–MS/MS	Acquity UPLC BEH C18	9.5	Extraction with acetonitrile containing 15% formic acid (*v*/*v*); purification by MCX sorbent (mixed-mode cation-eXchange sorbent)	70.5–119.2	[41]
Eggs	74 veterinary drugs	LOQ: 0.1–17.3	UPLC–MS/MS	Shimpack XR-ODS III column	16	Extraction with 1% acetic acid in acetonitrile; clean-up with Fe_3_O_4_-MWCNTs (magnetic multi-walled carbon nanotubes)	60.5–114.6	[22]
Livestock foods	155 veterinary drugs	LOD: 0.5–5	UHPLC–QTRAP–MS	Eclipse Plus C18	22	Extraction with acetonitrile/water/formic acid mixture (80:19.8:0.2); SPE (PRiME HLB)	46.4–120	[42]
Animal-derived food	120 drugs	LOD: 0.5–3.0	LC–MS/MS	Hypersil Gold C18	60	Extraction with 90% ACN (*v*/*v*); SPE (HLB)	/	[43]
Chicken muscle eggs	280 chemical hazards	LOD: 0.05–10; LOQ: 0.1–20	UHPLC–MS/MS	ACQUITY UPLC BEH C18	40	Extraction with 1% acetic acid in 85% acetonitrile; Purification: EMR-lipid	55.4–139	This study

**Table 2 foods-14-01660-t002:** Occurrence of chemical hazards in poultry muscle and egg samples.

Compounds	Poultry Eggs (*n* = 144)				Poultry Muscle (*n* = 67)			
	Number of Detected	Maximum (μg/kg)	Minimum (μg/kg)	Average (μg/kg) *	Number of Detected	Maximum (μg/kg)	Minimum (μg/kg)	Average (μg/kg) *
Trimethoprim	6	292	1.75	86.4	_	_	_	_
Sulfadiazine	3	638	0.67	401	2	1.4	1.19	1.29
Sulfamethizole	_	_	_	_	2	5.77	1.33	_
Sulfamonomethoxine	5	0.97	0.62	0.86	3	2.18	1.81	2.06
Sulfathinoxaline	1	15.9	_	_	1	1.69	_	_
Sulfadimidine	_	_	_	_	2	8.37	2.15	_
Sulfaphenazole	_	_	_	_	1	1.16	_	_
Sulfameter	1	3.95	_	_	_	_	_	_
Sulfisomidine	1	0.62	_	_	_	_	_	_
Nalidixic acid	2	1.09	0.48	_	3	21.4	8.65	16.8
Oxolinic acid	_	_	_	_	4	2.07	0.58	0.87
Ofloxacin	1	0.82	_	_	_	_	_	_
Enrofloxacin	5	1.96	0.35	0.65	_	_	_	_
Pazufloxacin	1	1.77	_	_	_	_	_	_
Ciprofloxacin	1	2.92	_	_	_	_	_	_
Salinomycin	_	_	_	_	2	0.77	0.77	_
Maduramicin	3	0.65	0.33	0.49	_	_	_	_
Guanabenz	2	13	2	_	3	21.9	1.37	14.4
Nicarbazin	3	17.7	0.14	6.96	5	6.64	1.2	3.92
Diclazuril	8	8.56	0.46	3.15	5	55	1.22	16.3
Toltrazuril	_	_	_	_	1	333	_	_
Azithromycin	_	_	_	_	2	56.1	24.5	_
Metronidazole	1	19.1	_	_	_	_	_	_
Ronidazole	1	2.13	_	_	_	_	_	_
Flubendazole	7	0.56	0.11	0.28	_	_	_	_
Albendazole sulfoxide	2	1.79	0.88	_	_	_	_	_
Fenbendazole sulfone	1	2.28	_	_	_	_	_	_
Florfenicol	3	115	3	59.4	_	_	_	_
Griseofulvin	_	_	_	_	1	0.59	_	_
Isoxsuprine	1	0.91	_	_	1	0.11	_	_
Chlorpromazine	2	0.33	0.24	_	_	_	_	_
Cortisone	_	_	_	_	34	23.4	1.02	7.62
Hydrocortisone	_	_	_	_	8	39.6	8.52	22.7
Betamethasone	_	_	_	_	2	0.91	0.55	_
Progesterone	89	211	3.53	54.5	13	38.6	2.22	9.4
Androstendione	29	7.65	0.11	3.42	_	_	_	_
Corticosterone	_	_	_	_	33	8.08	0.79	3.15
Testosterone	_	_	_	_	1	2.86	_	_
Basic violet 1	_	_	_	_	1	0.59	_	_
Rhodamine B	2	0.75	0.12	_	_	_	_	_
4-formylaminoantipyrine	1	2.2	_	_	_	_	_	_
Chlorpheniramine	4	1.62	0.72	1.17	_	_	_	_
Clomipramine	1	0.36	_	_	_	_	_	_

* Average of the detected samples.

## Data Availability

The original contributions presented in the study are included in the article/Appendix A; further inquiries can be directed to the corresponding author.

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
