# Peer review of "High-Throughput Determination of Multiclass Chemical Hazards in Poultry Muscles and Eggs Using UPLC–MS/MS"

_foods, 2025, doi:10.3390/foods14101660_

Round 1
Reviewer 1 Report
Comments and Suggestions for Authors
The presented study is dedicated to the determination of 282 hazardous chemicals of different classes (including sulfonamides, macrolides, pesticides, hormones, drugs, pigments, etc.) in poultry muscles and eggs using HPLC-MS/MS. For sample preparation authors used QuEChERS technique, including extraction with 1% formic acid-acetonitrile solution (15:85, v/v) and purification with EMR-lipid sorbent. The method was tested on real samples, and about 40 compounds were found.
I’ve got several comments requiring the authors’ attention.
1) Please, add letters A and B in Figure A1.
2) Lines 71-73: “282 chemical standards and 44 isotopic internal standards (IS) were purchased from TRC (Toronto, Canada) or Alta Technology Ltd (Tianjin, China), as listed in Appendix Table A1.” – As I see, Table A1 doesn’t contain information about manufacturer.
3) Section 2.1. Please, add information concerning ammonium fluoride, anhydrous sodium sulfate and sodium chloride. Also, did you use hydrochloric acid in this work?
4) Line 197-198: “In this experiment, the purification effects of five sorbents, 50 mg C18, 25 mg PSA, 50 mg Z-Sep/C18, 10 mg MWNTs, and 50 mg EMR-Lipid, were compared” – why did you use different amounts of sorbents?
5) Line 267: “The results, as presented in Table A2 in the Appendix” – there is no Table A2 in the Appendix.
6) Please, add chromatograms of blank samples and several real samples, for example, salted duck egg sample with 6 targets.
7) Line 308: “The linear range of the target was 0.2-500 μg/kg” – I suppose, it was 0.2-500 μg/L.
8) Abstract: “It was applied in the real samples detection, and 37 chemicals including metronidazole were found in 228 poultry samples with concentration range of 0.11-638 μg/kg.” – According to Table 1, there were 211 real samples analyzed (144 eggs + 67 muscles). Also, there are 44 chemicals in Table 1, and the concentration of Fipronil is 0.05 μg/kg, which is out of the range stated. Finally, why did you emphasize metronidazole?
9) Line 262-263: “Compared to the published literature focused on the multi-residues detection in eggs and muscles, this study presented comparable or higher senesitivity.” – Please compare the developed method with existing methods including matrix, number of compounds, analysis time and detection sensitivity. I suggest adding a comparison table.
10) Section 3.1.2. – I think you should rewrite this fragment. For example, you stated that “The organic phases were compared with methanol, acetonitrile, and methanol:acetonitrile (1:1, v/v)”. It seems like you had some organic phases, and they were compared with MeOH, AcN and MeOH:AcN. I suggest rewriting this as “The organic phases compared were methanol, acetonitrile, and methanol:acetonitrile (1:1, v/v)”. Also, you twice stated that MeOH, AcN and MeOH:AcN were compared – Lines 140-141 and 148-149.
11) According to table A1, some analytes with equal mass were almost or completely coeluted, for example, 2-18, 10-20, 12-17, 29-30, 125-131, 143-230, 164-165, 221-224. Also, compounds 209 and 222; 237 and 273 are the same one. Please, check m/z values for Testosterone-D2, it’s the same as for Testosterone.
12) Section 3.2 – You discussed the choice of extraction solvent and purification material, but why did you use NaCl and Na2SO4 in such amounts? Did you use some other work as basis for your sample preparation method?
13) Line 46-49: “Long-term consumption of products with excessive chemical residues may cause chronic toxicological effects, such as drug resistance, allergic reactions, and even teratogenic, carcinogenic, or mutagenic effects [2-4]” – Article [2] is devoted to distribution of strawberry plant metabolites, not hazardous chemicals, so it doesn’t fit here.
Please, check the manuscript thoroughly, I’ve found several typos, for example:
1) Line 100: “and 5 μL negative ion mode” – missed “in”.
2) Line 162: “. and” – dot instead of comma.
3) Line 193: “lipidss”.
4) Line 201: “was the seacod slection”.
5) Line 263: “senesitivity”.
Author Response
The presented study is dedicated to the determination of 280 hazardous chemicals of different classes (including sulfonamides, macrolides, pesticides, hormones, drugs, pigments, etc.) in poultry muscles and eggs using HPLC-MS/MS. For sample preparation authors used QuEChERS technique, including extraction with 1% formic acid-acetonitrile solution (15:85, v/v) and purification with EMR-lipid sorbent. The method was tested on real samples, and about 40 compounds were found.
I’ve got several comments requiring the authors’ attention.
1) Please, add letters A and B in Figure A1.
Response: Thank you for your suggestion. We have added A and B in Figure A1.
2) Lines 71-73: “282 chemical standards and 44 isotopic internal standards (IS) were purchased from TRC (Toronto, Canada) or Alta Technology Ltd (Tianjin, China), as listed in Appendix Table A1.” – As I see, Table A1 doesn’t contain information about manufacturer.
Response: Thank you for your comment. The sentence was revised to: "A total of 280 analytical standards and 44 isotope-labeled internal standards (listed in Appendix Table A1), all with purity >98%, were obtained from TRC (Toronto, Canada) or Alta Scientific Co., Ltd. (Tianjin, China). "
3) Section 2.1. Please, add information concerning ammonium fluoride, anhydrous sodium sulfate and sodium chloride. Also, did you use hydrochloric acid in this work?
Response: Thank you for your comments. We have supplemented the relevant information on ammonium fluoride, anhydrous sodium sulfate, and sodium chloride in the text with appropriate markings: "disodium EDTA (Na2EDTA), sodium chloride, anhydrous sodium sulfate, and ammonia were purchased from Sinopharm Chemical Reagent Co (Beijing, China); and ammonium fluoride (purity >99%) was purchased from Aladdin Biochemical Technology Co., Ltd. (Shanghai, China) "
Hydrochloric acid was not used in the experiments, that has been deleted from the manuscript.
- Line 197-198: “In this experiment, the purification effects of five sorbents, 50 mg C18, 25 mg PSA, 50 mg Z-Sep/C18, 10 mg MWNTs, and 50 mg EMR-Lipid, were compared” – why did you use different amounts of sorbents?
Response: The dosage of different sobents were determined upon the literature review or pre-experiment basis.:
PSA (25 mg): Referenced Frenich et al. (Anal. Chim. Acta 2010, 661, 150–160), where 25 mg PSA effectively purified 1 mL supernatant.
MWNTs (10 mg): Wu et al. (J. Chromatogr. B 2014, 965, 197–205) used 75 mg MWCNTs for 20 mL supernatant, while Xu et al. (Food Chem. 2019, 276, 419–426) applied 15 mg Fe3O4-MWCNTs for 2 mL extracts. A comparable dosage 10 mg was used for 1 mL supernatant in our study.
5) Line 267: “The results, as presented in Table A2 in the Appendix” – there is no Table A2 in the Appendix.
Response: This is our fault. Table A2 presented in the original version and was omitted before submission considering its limited information. Now the phase was deleted.
6) Please, add chromatograms of blank samples and several real samples, for example, salted duck egg sample with 6 targets.
Response: Thank you for your suggestion. The chromatograms of detected compounds in stardard soultion as well as in real sample were added as Figure 9 in the revised manuscript.
7) Line 308: “The linear range of the target was 0.2-500 μg/kg” – I suppose, it was 0.2-500 μg/L.
Response: We sincerely appreciate your careful identification of this error. We have revised the unit of concentration range to "μg/L" as suggested troughout the paper.
8) Abstract: “It was applied in the real samples detection, and 37 chemicals including metronidazole were found in 228 poultry samples with concentration range of 0.11-638 μg/kg.” – According to Table 1, there were 211 real samples analyzed (144 eggs + 67 muscles). Also, there are 44 chemicals in Table 1, and the concentration of Fipronil is 0.05 μg/kg, which is out of the range stated. Finally, why did you emphasize metronidazole?
Response: Thank you for your suggestion. This was indeed a typographical error. We have verified the sample numbers and detection results, and confirmed that fipronil was actually not detected. the text has been revised to:
Abstract: “It was applied in real sample detection, and 43 chemicals including metronidazole were found in 211 poultry samples, with a concentration range of 0.11-638 μg/kg.”
Metronidazole, a nitroimidazole-class antibiotic and antiprotozoal agent, is commonly used in veterinary medicine to treat anaerobic bacterial infections and coccidiosis. However, it has been identified as a high-risk drug for residue violations. Residual metronidazole in animal-derived food products may enter the human food chain, and long-term exposure has been associated with potential carcinogenic effects and antimicrobial resistance development. Due to these significant health concerns, China has explicitly listed metronidazole as a prohibited substance in food-producing animals. This is the reason why we particularly emphasize metronidazole.
” 9) Line 262-263: “Compared to the published literature focused on the multi-residues detection in eggs and muscles, this study presented comparable or higher senesitivity.” – Please compare the developed method with existing methods including matrix, number of compounds, analysis time and detection sensitivity. I suggest adding a comparison table.
Response: Thank you for your suggestion. A summary of the literature on multi-class veterinary drug residues in egg and chicken muscle matrices has been compiled in Table 1 in the revised manuscript.
10)Section 3.1.2. – I think you should rewrite this fragment. For example, you stated that “The organic phases were compared with methanol, acetonitrile, and methanol:acetonitrile (1:1, v/v)”. It seems like you had some organic phases, and they were compared with MeOH, AcN and MeOH:AcN. I suggest rewriting this as “The organic phases compared were methanol, acetonitrile, and methanol:acetonitrile (1:1, v/v)”. Also, you twice stated that MeOH, AcN and MeOH:AcN were compared – Lines 140-141 and 148-149.
Response: Thank you for your suggestions. We have revised this section as follows:”The mobile phase composition was systematically optimized to achieve optimal response and separation for all 280 target compounds. Three organic phase compositions were evaluated: methanol, acetonitrile, and methanol–acetonitrile (1:1, v/v). Concurrently, two aqueous phase formulations were compared for use in positive-ion mode: (1) 0.1% formic acid and (2) 0.5 mmol/L ammonium fluoride containing 0.1% formic acid. The results (Fig. 1A, 1B) demonstrated that the addition of 0.5 mmol/L ammonium fluoride into a 0.1% formic acid aqueous solution significantly improved chromatographic resolution and enhanced the mass spectrometric response intensities for most target analytes, compared to using a formic acid solution alone. In organic phase screening, the methanol–acetonitrile mixed system exhibited superior comprehensive performance (response intensity and resolution), markedly outperforming individual solvent systems, and was therefore selected as the optimal organic phase.
Based on these findings, the final optimized mobile phase combination for positive-ion mode analysis was established as follows: methanol–acetonitrile (1:1, v/v) with a 0.5 mmol/L ammonium fluoride aqueous solution containing 0.1% formic acid.”
11)According to table A1, some analytes with equal mass were almost or completely coeluted, for example, 2-18, 10-20, 12-17, 29-30, 125-131, 143-230, 164-165, 221-224. Also, compounds 209 and 222; 237 and 273 are the same one. Please, check m/z values for Testosterone-D2, it’s the same as for Testosterone.
Response: We sincerely apologize for the previous inaccuracies regarding the retention times for compounds. All duplicate compounds have been removed from the dataset to ensure data integrity. We have revised Appendix Table A1 by removing duplicate compounds and verified that the m/z values for Testosterone-D2 have been corrected to 291.
12) Section 3.2 – You discussed the choice of extraction solvent and purification material, but why did you use NaCl and Na2SO4 in such amounts? Did you use some other work as basis for your sample preparation method?
Response: The procedure was modified from Xu et al.'s study (2019), where the comparison between 4 g Na2SOâ‚„/1 g NaCl and 4 g MgSOâ‚„/1 g NaCl revealed chelation-induced recovery reduction for some targets. Consequently, Na2SO4 was selected over MgSO4.In our study, given the reduced sample size of 2 g, we proportionally decreased the desiccant amounts to 1 g Na2SOâ‚„and 1 g NaCl.
13) Line 46-49: “Long-term consumption of products with excessive chemical residues may cause chronic toxicological effects, such as drug resistance, allergic reactions, and even teratogenic, carcinogenic, or mutagenic effects [2-4]” – Article [2] is devoted to distribution of strawberry plant metabolites, not hazardous chemicals, so it doesn’t fit here.
Response: We thank the reviewer for noting this discrepancy. The reference has been removed in the revised manuscript.
Please, check the manuscript thoroughly, I’ve found several typos, for example:
1) Line 100: “and 5 μL negative ion mode” – missed “in”.
Response: Thank you for identifying the error. This has been corrected to 'and 5 μL in negative ion mode' and marked accordingly.
2) Line 162: “. and” – dot instead of comma.
Response: The erroneous punctuation marks in the text have been corrected.
3) Line 193: “lipidss”.
Response: The text has been revised to 'lipids' with the modification clearly marked.
4) Line 201: “was the seacod slection”.
Response: The spelling errors in the text have been corrected.
5)Line 263: “senesitivity”.
Response: The spelling has been corrected to 'sensitivity' with change tracking applied.
Reviewer 2 Report
Comments and Suggestions for Authors
April 8, 2025
Dear authors,
After reviewing the article titled “High-throughput determination of multi-class chemical hazards in poultry muscles and eggs using UPLC-MS/MS” which was submitted for possible publication in the Foods journal. I consider that the work addresses an interesting and ad hoc topic with the scope of the journal. However, I think that some aspects could help it improve.
Abstract
Do not use abbreviations without first stating their meaning at least once. For example, enhanced matrix removal (EMR). Apply to the whole document.
Introduction
The introduction is well written and includes relevant information about the topic, the research problem, the background and concludes by mentioning the objectives of the research. I consider it to be adequate. I have only a few minor suggestions regarding the drafting.
- Lines 55-57, change “LC-MS/MS with high selectivity, high sensitivity, and low detection limits is a common method for drug residue analysis of animal foods” by LC-MS/MS is a common method for analyzing drug residue in animal foods. It is characterized by high selectivity, sensitivity and low detection limits.
- Line 48 It is recommended that the types of contaminants likely to reach the final product be improved in order to provide the reader with a clear indication of which target analytes will be determined in the proposed method.
Materials and Methods
The purity of the standards must be declared.
Line 117, It is imperative to verify that all units are in accordance with the international system of units. It is imperative to note that the term 'litre' is written as 'L', not 'l'. This stipulation applies universally throughout the document.
It is imperative that the sample preparation subsection contains a reference. While a method is being proposed, it must be based on another article or a technical note from the supplier of the Quechers kit.
Since determinations are made at trace levels, it is incumbent upon the authors to employ Quality Assurance of the Method. It is imperative to incorporate a new subsection within the Materials and Methods section, wherein this critical information is delineated.
It is understood from the abstract that the method was validated. Consequently, a validation section specifying the evaluated parameters and the guide adhered to is required.
It is incumbent upon authors to indicate whether statistical analyses have been performed.
It is considered that instrumental conditions should follow sample preparation so that there is a logical sequence in the process.
Results and discussion
Line 195 change lipidss by lipids.
Line 201 change resluts by results.
In Section 3.2.2, the authors are required to indicate whether the recoveries obtained are in accordance with any existing validation guide. Furthermore, they are required to comment on the matrix effect observed in each of the absorbent materials that were tested.
The selection criterion for the compounds presented in Figures 4 and 6 is unclear.
3.3.2 Linear range and method limit of quantification (LOQ) It is imperative that the detection limit (LOD) of each analyte is declared, in addition to the linear range of each analyte for both muscle and egg. In this regard, the units for muscle should be microgram per kilogram. It is important to note that the authors' demonstration of equipment linearity in the supplemental material does not equate to method linearity. It is therefore imperative to rectify this misapprehension or to provide the requested information.
Line 265 change senesitivity by sensitivity.
Lines 264-265 The authors state "Compared to the published literature focused on the multi-residues detection in eggs and muscles, this study presented comparable or higher sensitivity". Reference the literature with which it was compared to arrive at this statement.
Line 268 How were the muscle and egg blanks prepared or obtained?
Table A2 is absent in the Appendix of the supplemental material.
Line 294 change risksd by risk.
I consider that a subsection called limitations, and future perspectives should be added to the results and discussion section.
Conclusion
It is important to note that the units of the linear range are expressed in ug/kg, whereas in the supplementary material, they are indicated in the linearity of the equipment, i.e. ug/L. This discrepancy necessitates further elucidation.
It is imperative to note that the conclusions section should not contain references, given that the conclusions are based on the research carried out. For this reason, it is recommended that the section containing the conclusions be revised to remove any references to specific sources.
References
It is imperative to verify that all references are formatted consistently and adhere to the guidelines outlined in the journal's guide author. It should be noted that the name of the journal may be displayed in full or in abbreviated form.
The reviewer considers that the manuscript may be suitable for publication following major revisions. Considering a language review by a professional in the area or a native speaker
King regards
Comments on the Quality of English LanguageIt is evident that there are a number of typographical and syntax errors. It is therefore recommended that a review of the language is undertaken.
Author Response
Comments and Suggestions for Authors
April 8, 2025
Dear authors,
After reviewing the article titled “High-throughput determination of multi-class chemical hazards in poultry muscles and eggs using UPLC-MS/MS” which was submitted for possible publication in the Foods journal. I consider that the work addresses an interesting and ad hoc topic with the scope of the journal. However, I think that some aspects could help it improve.
Abstract
(1)Do not use abbreviations without first stating their meaning at least once. For example, enhanced matrix removal (EMR). Apply to the whole document.
Response: Thank you for your suggestion. The manuscript has been revised to include the complete English nomenclature.
Introduction
The introduction is well written and includes relevant information about the topic, the research problem, the background and concludes by mentioning the objectives of the research. I consider it to be adequate. I have only a few minor suggestions regarding the drafting.
(2)Lines 55-57, change “LC-MS/MS with high selectivity, high sensitivity, and low detection limits is a common method for drug residue analysis of animal foods” by LC-MS/MS is a common method for analyzing drug residue in animal foods. It is characterized by high selectivity, sensitivity and low detection limits.
Response: Thank you for your suggestion. We have revised the sentence as follows:
" LC-MS/MS is a common method for analyzing drug residues in animal-derived foods. It is characterized by high selectivity and sensitivity and low detection limits.”
(3)Line 48 It is recommended that the types of contaminants likely to reach the final product be improved in order to provide the reader with a clear indication of which target analytes will be determined in the proposed method.
Response: Following your recommendation, we have specified representative analyte categories in final products (including sulfonamides, quinolones, tetracyclines, and β-agonists) to better define the method's target compounds.
Materials and Methods
(4)The purity of the standards must be declared.
Response: Thank you for your suggestions. We have revised the sentence as follows:
“A total of 280 analytical standards and 44 isotope-labeled internal standards (listed in Appendix Table A1), all with purity >98%, were obtained from TRC (Toronto, Canada) or Alta Scientific Co., Ltd. (Tianjin, China).”
(5)Line 117, It is imperative to verify that all units are in accordance with the international system of units. It is imperative to note that the term 'litre' is written as 'L', not 'l'. This stipulation applies universally throughout the document.
Response: Thank you for your comments. We have carefully reviewed the entire manuscript and made the necessary corrections by changing "l" to "L" with appropriate markings throughout the text.
(6)It is imperative that the sample preparation subsection contains a reference. While a method is being proposed, it must be based on another article or a technical note from the supplier of the Quechers kit.
Response: Thank you for your suggestions. Relevant references have now been cited in the sample preparation section.
(7)Since determinations are made at trace levels, it is incumbent upon the authors to employ Quality Assurance of the Method. It is imperative to incorporate a new subsection within the Materials and Methods section, wherein this critical information is delineated.
Response: Thank you for your suggestion. We have added a new section (2.6) to address this point in the manuscript.
“2.6 Quality assurance
A robust quality assurance design for LC-MS/MS ensures data integrity, regulatory compliance, and reliable results. In this study, the method was validated according to China National Standard GB/T 27417-2017 Conformity assessment—Guidance on validation and verification of chemical analytical methods. Various parameters including linearity, accuracy, precision, limit of detection (LOD), and limit of quantification (LOQ) were evaluated during the course.
The performance of the LC-MS/MS analytical system was verified using the evaluated standard solutions, including the analytes and internal standards. With the calibration curve established, accuracy should be +15% for all points except the lowest calibrator (+20%). In addition, the curve slope should be r2 > 0.995. A spiked blank matrix sample was used as the quality control (QC) sample due to the lack of certified reference materials. A blank sample, as well as QC samples, was applied for every 20 injections in a batch analysis.”
(8)It is understood from the abstract that the method was validated. Consequently, a validation section specifying the evaluated parameters and the guide adhered to is required.
Response: Thank you for your suggestions. The following sentence has been added to the Method section: “The method was validated in compliance with the requirements of China National Standard GB/T 27417-2017 Conformity assessment—Guidance on validation and verification of chemical analytical methods.”
(9)It is incumbent upon authors to indicate whether statistical analyses have been performed.
Response: This study primarily focuses on the determination of sensitivity (LOD/LOQ), reproducibility (RSD%), and accuracy (recovery), and therefore did not employ statistical analysis.
(10)It is considered that instrumental conditions should follow sample preparation so that there is a logical sequence in the process.
Response: Thank you for your suggestion. The sample pretreatment section has been moved before the instrument conditions and has been labeled accordingly.
Results and discussion
(11)Line 195 change lipidss by lipids.
Response: Thank you for your comments. The indicated spelling mistakes have been rectified and properly annotated in the text.
(12)Line 201 change resluts by results.
Response: Thank you for your comments. The indicated spelling mistakes have been rectified and properly annotated in the text.
(13)In Section 3.2.2, the authors are required to indicate whether the recoveries obtained are in accordance with any existing validation guide. Furthermore, they are required to comment on the matrix effect observed in each of the absorbent materials that were tested.
Response: The relevant content has been added to Section 3.2.2 and marked accordingly in the revised manuscript.
(14)The selection criterion for the compounds presented in Figures 4 and 6 is unclear.
Response: Figures 5 and 6 present statistical comparisons of target compounds demonstrating >20% recovery differences between the two optimal purification sorbents (EMR-lipid and PSA) in chicken muscle and (EMR-lipid and Z-Sep/C18) egg matrices.
(15)3.3.2 Linear range and method limit of quantification (LOQ) It is imperative that the detection limit (LOD) of each analyte is declared, in addition to the linear range of each analyte for both muscle and egg. In this regard, the units for muscle should be microgram per kilogram. It is important to note that the authors' demonstration of equipment linearity in the supplemental material does not equate to method linearity. It is therefore imperative to rectify this misapprehension or to provide the requested information.
Response: The detection limit LOD) of the method has been provided in Table A1 , along with the linear ranges for egg and chicken meat matrices. Following reviewer 1’s comment, the unit of linear range was μg/L.
(16) Line 265 change senesitivity by sensitivity.
Response: The spelling has been corrected to 'sensitivity' with change tracking applied.
(17)Lines 264-265 The authors state "Compared to the published literature focused on the multi-residues detection in eggs and muscles, this study presented comparable or higher sensitivity". Reference the literature with which it was compared to arrive at this statement.
Response: We sincerely appreciate your suggestions. A summary of the literature on multi-class veterinary drug residues in egg and chicken muscle matrices has been compiled in Table 1.
(18)Line 268 How were the muscle and egg blanks prepared or obtained?
Response: The sample preparation section has been supplemented with the following details:
Blank samples of chicken muscle and eggs used for method development or validation were kindly donated by China Agricultural University.
Poultry muscle samples were prepared by removing connective tissues, dicing into small pieces , and homogenizing using a mechanical grinder. Egg samples were prepared by pooling and homogenizing 10 whole eggs.
(19)Table A2 is absent in the Appendix of the supplemental material.
Response: We appreciate you bringing this to our attention. Table A2 has now been deleted from the text.
(20)Line 294 change risksd by risk.
Response: Thank you for identifying these spelling inaccuracies. The text has been amended to: risk.
(21)I consider that a subsection called limitations, and future perspectives should be added to the results and discussion section.
Response: The following statement has been appended to conclude the manuscript:”While this method enables broad compound coverage, certain analytes demonstrate limited recovery efficiency, and the reliance on costly EMR-lipid sorbents may hinder large-scale implementation. Future studies should prioritize advanced sorbent development to improve recovery and automation-integrated workflows to boost throughput.”
Conclusion
(22)It is important to note that the units of the linear range are expressed in ug/kg, whereas in the supplementary material, they are indicated in the linearity of the equipment, i.e. ug/L. This discrepancy necessitates further elucidation.
Response: We apologies for this oversight. The units have been revised to 'μg/L' throughout the paper.
(23)It is imperative to note that the conclusions section should not contain references, given that the conclusions are based on the research carried out. For this reason, it is recommended that the section containing the conclusions be revised to remove any references to specific sources.
Response: Thank you for your suggestion. The references in the conclusion section have been removed.
References
(24)It is imperative to verify that all references are formatted consistently and adhere to the guidelines outlined in the journal's guide author. It should be noted that the name of the journal may be displayed in full or in abbreviated form.
Response: Thank you for your suggestion. The reference format has been adjusted according to the journal's requirements.
The reviewer considers that the manuscript may be suitable for publication following major revisions. Considering a language review by a professional in the area or a native speaker
King regards
(25)Comments on the Quality of English Language
It is evident that there are a number of typographical and syntax errors. It is therefore recommended that a review of the language is undertaken.
Response: Thank you for your comment. The revised manuscript has been reviewed by the MDPI's language editing service.